# Goal Reduction with Loop-Removal Accelerates RL and Models Human Brain Activity in Goal-Directed Learning

**Huzi Cheng**
Department of Psychological and Brain Sciences
Indiana University Bloomington
hzcheng15@gmail.com

**Joshua W. Brown**
Department of Psychological and Brain Sciences
Indiana University Bloomington
jwmbrown@iu.edu

## Abstract

Goal-directed planning presents a challenge for classical RL algorithms due to the vastness of the combinatorial state and goal spaces, while humans and animals adapt to complex environments, especially with diverse, non-stationary objectives, often employing intermediate goals for long-horizon tasks. Here, we propose a goal reduction mechanism for effectively deriving subgoals from arbitrary and distant original goals, using a novel loop-removal technique.[1] The product of the method, called *goal-reducer*, distills high-quality subgoals from a replay buffer, all without the need for prior global environmental knowledge. Simulations show that the *goal-reducer* can be integrated into RL frameworks like Deep Q-learning and Soft Actor-Critic. It accelerates performance in both discrete and continuous action space tasks, such as grid world navigation and robotic arm manipulation, relative to the corresponding standard RL models. Moreover, the *goal-reducer*, when combined with a local policy, without iterative training, outperforms its integrated deep RL counterparts in solving a navigation task. This goal reduction mechanism also models human problem-solving. Comparing the model's performance and activation with human behavior and fMRI data in a treasure hunting task, we found matching representational patterns between a *goal-reducer* agent's components and corresponding human brain areas, particularly the vmPFC and basal ganglia. The results suggest that humans may use a similar computational framework for goal-directed behaviors.

## 1 Introduction

Humans and animals must develop capabilities to pursue time-varying goals in continuously changing environments. The pressure for survival prohibits slow, linear adaptation to different goals, i.e., learning value functions from scratch for each new objective. A quick and versatile paradigm is necessary for such goal-directed learning scenarios. However, traditional Reinforcement Learning (RL) algorithms are not specifically designed for this, encountering challenges in the goal-directed learning context. They are highly optimized for scenarios with relatively fixed goals, e.g., winning in Go, or reducing building energy consumption [51]. In these situations, iterative methods like the

---

[1]Our code is available at github.com/chenghuzi/goal-reducer.

Bellman equation are effective for approximating value/advantage/negative cost functions across various states, maintaining stability over time. Nonetheless, if the goal changes during training, classical RL becomes highly inefficient due to: 1) The significant increase in state space caused by the introduction of the goal set; 2) The inability to reuse experiences across different goals.

On the other hand, traditional heuristic algorithms like Dijkstra's shortest path algorithm [12] and the A* path-finding algorithm [24] excel in these problems. These algorithms leverage goal-independent environment knowledge to construct real-time goal-conditioned maps during navigation. They use selective intermediate goal states to simplify the problem, thus reducing pathfinding time. By breaking down a distant goal into nearer ones, these algorithms utilize local knowledge, independent of goal changes, to solve the problem. More importantly, this principle resembles ways humans use to deal with complex problems. Recently, a number of studies have shown that, besides stimuli and association representation, human and animal brains leverage goal and reduced subgoal representations to solve tasks in various settings [44, 11, 45, 40]. But a common drawback of these algorithms is that they require predefined representations of all states in the task, prohibiting them from scaling to large-scale realistic tasks.

In this paper, we try to bridge such goal reduction mechanisms with neural models. The resulting algorithm is an effective *goal-reducer* that can accelerate and beat standard RL in different tasks through recursively reducing complex goals into simpler subgoals. The main contributions of this paper are two-fold: **1)** Computationally, we propose novel methods to train an effective *goal-reducer*. After just random walking, it extracts nearly optimal subgoals when the state-goal combination space is large without prior knowledge about the cost/effort between them. This is based on the effective representation structure learned during training. This mechanism, as shown below, can not only be used to accelerate standard RL algorithms in various settings but also to guide navigation in a multi-goal setting without iterative Bellman equation convergence processes and outperforms RL accelerated with it, let alone standard RL itself. **2)** Biologically, by comparing the model's activity with brain fMRI data in the same cognitive task, we show the similarity between the model's component, e.g., the *goal-reducer*, and brain regions like vmPFC and putamen, highlighting the potential of using this model to explain how brains solve goal-directed problems.

## 2 Related work

### 2.1 Goal-directed RL in deep learning

Developing an effective goal-directed algorithm has been a long-lasting open question in the RL community [2]. Various methods have been proposed to mitigate the experience sparsity issue in goal-directed learning. For an extensive review, we refer the reader to [35]. Here, we introduce some representative solutions, where some focus on the modification of reward functions, using tricks like average action cost [25] and sibling trajectory average goals [49] to alleviate the sharpness in reward distribution in the early stage of training. Another line of work uses planning to solve the problem. For example, an explicit graph can be constructed from sensory inputs for traditional path-finding algorithms [16, 55, 34, 26]. However, we see these approaches as not completely neural-based models, and the pathfinding algorithm part may prohibit them from smoothly scaling to tasks with larger state spaces.

Another family of algorithms uses subgoal generation as a core mechanism to resolve the same issue. [18] trained a GAN to predict intermediate goals. There are also attempts to construct subgoals with different heuristics: [56] uses the uncertainty of Q-functions to help train subgoals. [9] treats the midpoint of value functions as optimal subgoals during training. [3] optimizes subgoal generation by minimizing integrated segment costs produced by "local" subgoals. These subgoal-related approaches are intriguing as they match the "divide and conquer" principle in a neural way, but a common limitation of these methods is that they work in a bootstrapping manner, i.e., the quality of the Q-function indirectly determines the quality of subgoals generated, as it is involved in the subgoal sampling process.

### 2.2 Goal-directed learning and subgoal generation in neuroscience

There exists a duality of interest in neuroscience on goal-directed learning that can be dated back to the 1940s, when [48] famously showed that rats can take shortcuts that hadn't been experienced

before to reach goals in a maze. Later research showed that there are representations for different goals in the brain that are tailored for planning optimal paths to the ultimate goal [13]. Models have been built for these processes to reveal possible mechanisms of how the brain may make use of these goal representations to calculate subgoals (or called "landmarks") for navigation [46, 47]. Recently, more empirical and modeling work has come out showing that the subgoal navigation hypothesis, and the underlying cognitive map theory that supports it [5], could be implemented by animals [23, 45] and humans [50, 15, 54]. Altogether, these studies imply the existence of a subgoal generation mechanism in the brain to support *effective* planning in complex environments.

However, many models for subgoal generation in neuroscience still either rely on manual/one-hot coding of states [15, 54] or focus on revealing the existence rather than the potential development of subgoals during training [45]. We therefore think that, in these two fields, the community of deep RL and neuroscience, there is space for a neurally plausible algorithm that can provide a biological model of how subgoals are naturally generated from the brain during training and can show its computational efficiency over plain RL algorithms.

Our resulting work presents a trained *goal-reducer* neural network that generates nearly optimal subgoals without requiring additional value information from the environment. This approach distinguishes itself from other subgoal generators that depend on Q-functions. It can integrate seamlessly with standard RL frameworks and independently operate with a local policy that just learns associations in neighboring states. Using the latter approach, we also demonstrate its capability in solving cognitive tasks in a human-like manner and its correspondence with various brain regions, indicating its potential for modeling human problem-solving processes.

## 3 Methods

### 3.1 Problem formulation

For a goal-directed Markov decision process, we characterize it with $(\mathcal{S}, \mathcal{A}, \mathcal{G})$, where $\mathcal{S}$ and $\mathcal{A}$ are the observation and action spaces, respectively, and $\mathcal{G}$ is the goal space. One interaction step in this environment can be written as $(s_t, g, a_t, r_t, \hat{g}_t, s_{t+1})$, where $s_t$ is the current observation, $g$ is the assigned goal, $a_t$ is the action executed, $r_t$ is the reward, $\hat{g}_t$ is the achieved goal, and $s_{t+1}$ is the next state. In some cases, $\mathcal{G} = \mathcal{S}$ occurs when a possible goal is among one of all states, e.g., spatial navigation when the input and goal space is all plausible locations. In other words, the goal and state space are the same. In such settings, we may use $\mathcal{G}$ and $\mathcal{S}$ interchangeably, and the interaction can be reduced to $(s_t, g, a_t, r_t, s_{t+1})$. But in some real-world tasks, $\mathcal{G}$ may be in a different space from $\mathcal{S}$: $\mathcal{G}$ could be a target coordinate in allocentric space when $\mathcal{S}$ is the pose of a multi-joint robot arm.

With this formulation, we first consider the simple case when $\mathcal{G} = \mathcal{S}$ as the general case can be easily extended from it. In goal-directed learning, $g \in \mathcal{G}$ may change substantially, making conventional RL algorithms inadequate due to the enormous size of $\mathcal{S} \times \mathcal{G}$. Additionally, in many scenarios, the reward is sparse, and the agent only receives a positive reward or avoids punishment when it reaches the goal state, i.e., when $\mathcal{G} = \mathcal{S}$ and $r(s_t, a_t, g) = \mathbb{1}(s_t = g)$. This, combined with the complexity of the state space, further complicates goal-directed learning.

We propose that, given a specific $g$, if an agent can effectively reduce a goal $g$ into a subgoal $s_g \in \mathcal{G}$ that is "closer" to its current state $s_t$, it may alleviate the task's difficulty. Furthermore, this can be applied **recursively** to find subgoals that are arbitrarily close to the current state $s_t$.

### 3.2 Effective *goal-reducer* through *Loop-removal sampling*

The most straightforward solution to the problem above is to train a function $\Phi$ that generates $s_g$, which we refer to as a *goal-reducer*: $\Phi(s_t, g) : \mathcal{S} \times \mathcal{G} \to \mathcal{G}$. The *goal-reducer* can reduce the computational burden of a policy $\pi(a|s, g)$ as it can now generate a subgoal $s_g = \Phi(s, g)$ for a hard task, under a basic assumption that harder problems can be decomposed into simpler problems as long as these problems are in the optimal path towards the final solution (Fig. 1A). Training such a *goal-reducer*, however, can be challenging. There are two intuitive strategies. First, one can sample $s_g$ uniformly from $\mathcal{S}$, as done by [9]. This approach is referred to as *Random sampling* (Fig. 1B left). An alternative is to sample $s_g$ from past experiences, a strategy known as *Trajectory sampling*. Assuming a sequence of interactions $(s_t, g, a_t, r_t, s_{t+1})$ for $t = 1, 2, \cdots, T$ is stored in memory, $s_g$ can be sampled from the $s_t$ in this sequence (Fig. 1B middle). A common technique in goal-directed

RL, Hindsight Experience Replay (HER), proposed by [2], conceptually resembles this approach, as it encourages the agent to learn associations between current states and some of its future states in the same trajectory. Other studies, such as [41], also find trajectory-based sampling effective in goal-directed learning, though expert experience is not required in our case [41]. Furthermore, neuroscience research indicates that episodic memories, i.e., sequences of experienced states, are vital for learning [20].

However, this strategy does not guarantee the sampled $s_g$ to be effective: an agent with limited environmental knowledge may simply engage in a random walk within the state transition graph, rendering the states experienced in the episodic memory ineffective for connecting the trajectory's start and end points. To address this, we introduce a third strategy, termed *Loop-removal sampling* (Fig. 1B right), which may mitigate this issue. The underlying rationale is that when the agent has minimal knowledge of the environment, the trajectories it creates will likely be random and involve numerous "loops". A "loop" occurs when the agent revisits the same state at least twice within a trajectory. The *Loop-removal sampling* posits that by eliminating these "loops" from the episodic memory, the $s_g$ sampled from the remaining trajectories will be more advantageous, as ineffective experiences are excluded, potentially resulting in a trajectory that more closely resembles a linear or shortest path in the best-case scenario.

While *Loop-removal sampling* is effective when $\mathcal{S}$ is discrete, i.e., observations are either the same or different, it faces challenges in environments where $s_i \in \mathcal{S}$ can be infinitely close but not the same to each other, e.g., when the observation is an image input for an agent's current location. This issue is also encountered in earlier concepts like [28]. To overcome this, we adopt an idea from persistent homology [14], defining a filtration process to determine the existence of "loops."

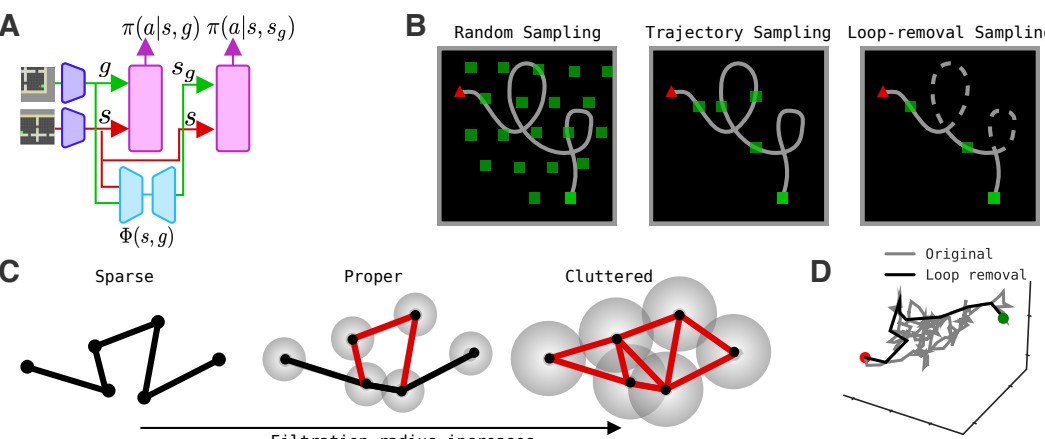

Fig. 1: **A**: A schematic view of how a *goal-reducer* can be integrated with a policy network to generate actions. **B**: 3 types of subgoal sampling strategies. Red triangles: current states, green squares: goals, light green squares: subgoals sampled. **C**: An diagram of filtration radius changes trajectory connectivity. **D**: An example of proper filtered trajectory (black) compared with original random walk (gray) in 3D space.

This process is schematically depicted in Fig. 1C: In a continuous space, trajectories are sparse, as distances between states are always greater than zero, making exact state overlaps unlikely. However, by assigning a filtration radius $\rho$ to each state and incrementally increasing it, the connectivity of the episodic memory trajectory changes. At a certain point, the algorithm detects a "loop" (proper case in Fig. 1C), where *Loop-removal sampling* is most effective. If the filtration radius continues to increase, the trajectory becomes fully interconnected, leading to *Loop-removal sampling* failure. We demonstrate that with an appropriate filtration parameter, a random walk path in 3D space can be effectively simplified (Fig. 1D) with $\rho$ set to 0.8. The optimal radius is determined through a grid search between 0 and the longest distance between any pair of points in the space, with the objective of maximizing the rate at which task performance improves with learning. This search process is also used in later experiments to maximize learning efficiency in terms of two efficiency metrics, *Optimality* and *Equidex* (described in detail below).

This approach may seem akin to the *Search on the Replay Buffer* by [17], but there is a key distinction: *Search on the Replay Buffer* depends on a value function to estimate distances among states, whereas *Loop-removal sampling* operates independently of any prior environmental knowledge. Instead, it relies on a minimal assumption applicable to all state spaces: loops indicate redundancy. Following this, we developed an online mechanism ( Algorithm 1 in appendix) to train the *goal-reducer*, $\Phi$, using loop-removed trajectories stored in $\mathcal{D}'$, which is refined from the replay buffer $\mathcal{D}$.

To test if a *goal-reducer* trained with *Loop-removal sampling* generates better subgoals compared to *Trajectory sampling* and *Random sampling*, we developed two metrics to quantify the quality of generated subgoals: *Optimality* and *Equidex*. First, we represent the effort required for an agent to reach goal $g$ from state $s$ as $||s, g||$, the distance across legal transitions of the state graph. This concept mirrors the idea of shortest distances, though it may not be symmetrical, i.e., $||s, g|| \neq ||g, s||$, due to potentially irreversible state transitions. Also, the state space graph of legal transitions may not be fully connected – because barriers may exist in the state space so that some transitions are not possible, $s_g$ is generally not simply the linear midpoint between $s$ and $g \in \mathcal{S}$. Building on this, *Optimality* is defined as:

$$Optimality(s, g, s_g) = ||s, g||/(||s, s_g|| + ||s_g, g||). \tag{1}$$

When *Optimality* $\to 1$, the total effort expended by an agent to reach $g$ via $s_g$ is near the optimal effort. However, a $s_g$ with an *Optimality* close to 1 may not be informative if $s_g$ is set to either $g$ or $s$, where no new information is provided. To address this, *Equidex* is introduced:

$$Equidex(s, g, s_g) = (||s_g, g|| - ||s, s_g||)/(||s, s_g|| + ||s_g, g||). \tag{2}$$

When *Equidex* $\to 0$, it indicates that the effort from $s$ to subgoal $s_g$ is similar to the effort from $s_g$ to the final goal $g$. This suggests that $s_g$ is situated on the hyperplane formed by midpoints in the state space between $s$ and $g$. Accordingly, as *Equidex* $\to 1$, the subgoal becomes closer to the current state, while a value nearing -1 suggests the subgoal is closer to the ultimate goal. In summary, the quality of a single subgoal can be characterized by both *Optimality* and *Equidex*. The ideal subgoal is one where *Optimality* equals 1 and *Equidex* equals 0.

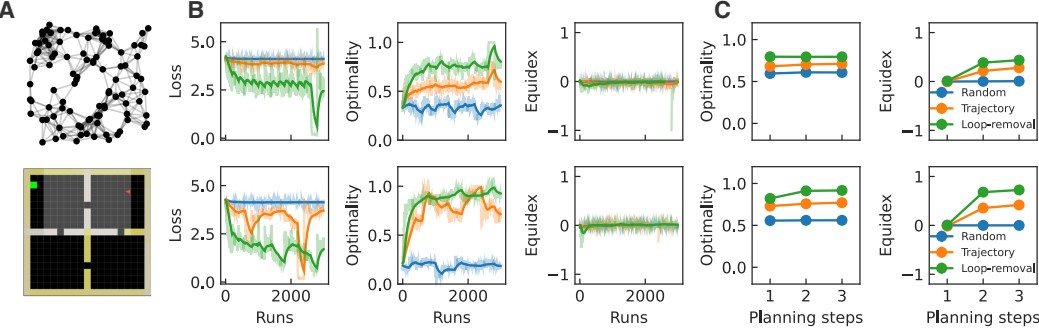

Fig. 2: *goal-reducer* training results of geometric random graph (top) and the four-room gridworld task (bottom) with different strategies. **A**: Environment examples. **B**: Left: training loss, middle: training *Optimality*, right: training *Equidex*. **C**: Left: *Optimality* change when applying a trained *goal-reducer* recursively, right: same, but for *Equidex*.

Using these indices, we assess the performance of the *goal-reducer* on two datasets. The first dataset derives from a constructed state graph, characterized by random connections among states and allowing self-connections (Fig. 2A top). We execute undirected random walks on this graph. The second is based on a four-room gridworld task, commonly used in multi-goal RL benchmark tests (Fig. 2A bottom). In this dataset, we simulate an agent exploring the environment without any prior knowledge of its structure.

For both datasets, we use the *goal-reducer* architecture to learn the $s, g \to s_g$ association, employing an VAE [31]. This network accepts concatenated representations of $s$ and $g$ as inputs, initially producing a latent probabilistic representation $z$ through the encoder Enc$_\Phi$. A prediction of $s_g$ is then generated by Dec$_\Phi$. To circumvent sparse and discrete encoding in both datasets, and also to avoid introducing prior knowledge, random embeddings are applied for each. Consequently, $s$,

$g$, and $s_g$ are represented as high-dimensional random vectors. All random embeddings remain fixed during training. The adopted loss function mirrors that of a classical VAE, comprising a subgoal reconstruction loss and a weighted ($\alpha$) KL divergence penalty with a prior latent distribution $p(z) \sim \mathcal{N}(0, I)$:

$$L = -\alpha D_{KL}(\text{Enc}_{\Phi}(s_g|s, g)||p(z)) + \mathbb{E}_{\text{Enc}_{\Phi}(z|s,g)}[\log \text{Dec}_{\Phi}(s_g|z)]. \tag{3}$$

Training results (Fig. 2B left) indicate that in both datasets *Loop-removal sampling* surpasses and is more stable than *Trajectory sampling*, which in turn outperforms *Random sampling* in terms of loss. The *Optimality* values follow a similar trend (Fig. 2B middle):

*Loop-removal sampling > Trajectory sampling ≫ Random sampling.*

Given that all subgoals are uniformly sampled from the dataset, the *Equidex* for all strategies approximates 0 (Fig. 2B right). To further examine the capability and stability of the trained *goal-reducer*, we assess its proficiency in recursive subgoal reduction. In this process, we input the predicted subgoal embedding into the *goal-reducer*, treating it as $g$ to generate a subsequent subgoal $s_g'$. By iterating this procedure for several steps, we postulate that the *goal-reducer*, particularly when trained with *Loop-removal sampling*, will produce subgoals that are both effective and sufficiently close to the current states to facilitate easier navigation. The experimental results reveal that, after three iterations of recursive goal reduction ($t = 3$, in Fig. 2C), the *goal-reducer* trained with *Loop-removal sampling* achieves the most favorable optimality distribution (Fig. 2C left).

In terms of *Equidex*, as illustrated in the right column of Fig. 2C, recursive goal reductions enhance the *goal-reducer*'s ability to predict subgoals that are nearer to the current states. This improvement is evidenced by the shift in the equidex distribution from $t = 1$ to $t = 3$. Together, the results in Fig. 2 demonstrate the superiority of *Loop-removal sampling* as a training strategy for *goal-reducer* to produce proper subgoals.

### 3.3 *goal-reducer* integrated with RL

The above experiments have shown the superiority of *Loop-removal sampling* over other strategies in training *goal-reducer* from environments without external knowledge about the "distance" between any two states in them. Next we integrate this process with RL algorithms into tasks with discrete and continuous action spaces using Deep Q-learning (DQL) [38] and Soft Actor-Critic (SAC) [22].

**Discrete case** In the discrete setting, DQL is used. The algorithm is trained to optimize a value function $Q$ through Bellman equation convergence:

$$Q_{\phi_{k+1}} = \arg\min_{\phi} \frac{1}{2} \mathbb{E}_{(s_t, g, a_t, s_{t+1}) \sim D} [Q_t^* - Q_\phi(s_t, g, a_t)]^2 \tag{4}$$

$$Q_t^* = r(s_t, g, a_t) + \gamma \mathbb{E}_{a_{t+1} \sim \pi(\cdot|s_{t+1}, g)} Q_{\phi_k}(s_{t+1}, g, a_{t+1}). \tag{5}$$

Since we assume *goal-reducer*, $\Phi(s, g)$, can learn to generate subgoals without a mature $Q$, we use it to accelerate the convergence of Eq. 4 by updating $Q$'s parameter using an extra regularization loss

$$L_{s_g} = \sum_{s_t, g} w_{s_t, g} \cdot D_{KL}\left[\pi(a_t|s_t, g) \middle\| \pi(a_t|s_t, \Phi(s, g))\right], \tag{6}$$

where the policy $\pi(a_t|s_t, g)$ is a categorical distribution $\text{softmax}Q(s_t, g)$ and the loss is weighted by the entropy of policy, $H[\pi(a_t|s_t, g)]$:

$$w_{s_t, g} = \begin{cases} 1, & \text{if } H[\pi(a_t|s_t, g)] \geq H[\pi(a_t|s_t, \Phi(s, g))] \\ 0, & \text{otherwise.} \end{cases} \tag{7}$$

**Continuous case** In continuous action space, SAC is used. Since $\pi(a_t|s_t, g)$ cannot be calculated explicitly, we approximate it with an online sampling process of actions executed $a_t'$:

$$L_{s_g} = \sum_{s_t, a_t', g} w_{s_t, g} \cdot \pi(a_t'|s_t, g) \cdot \log \pi(a_t'|s_t, \Phi(s, g)), \tag{8}$$

where the $w_{s_t,g} = 1$ if $\pi\big(a'_t|s_t, g\big) < \pi\big(a'_t|s_t, \Phi(s, g)\big)$ and otherwise 0. In both cases, $w_{s_t,g} = 1$ means $Q$ is more uncertain about the ultimate goal when compared with a trustworthy goal generated by the *goal-reducer*.

For both cases, for baseline RL methods (plain DQL and SAC) and their *goal-reducer* augmented version, to accelerate learning, we used Hindsight Experience Replay (HER) during training, a standard technique used to improve RL algorithm's performance in goal-directed learning [2].

### 3.4 Standalone *goal-reducer*

As shown in previous *Optimality* and *Equidex* experiments, *goal-reducer* can gradually reduce the "distance" between the agent and the goal through recursive goal reduction. We thus test if this mechanism alone can solve some tasks that are usually handled with model-free RL using Bellman equation iteration. To do this, we take an unsupervised approach: When an agent is initialized, it explores the environment randomly. We train the *goal-reducer* using such exploration trajectories. At the same time, we train a local policy $\pi_{\text{local}}\big(a_t|s_t, g\big)$ that only learns goals that are one step away from $s_t$ and generates a uniform distribution otherwise. When these two components are trained, we execute the planning process by detecting reachable goals using the entropy of $\pi_{\text{local}}$ (for details, see the appendix).

## 4 Results

### 4.1 *goal-reducer* accelerates standard RL

**Four-room maze navigation task**    In a modified mini-grid environment [10] (Fig. 3A left), an agent receives two images as inputs and outputs an action indicating which direction to go among four possible directions. One image is the partial observation in a four-room grid world maze, $s_t$, while the other image is a similar "picture" from the "goal" location, $g$. No other visual cue beyond the "picture" of the goal location is given, preventing the agent from cheating using easy visual landmarks like the green dots used in the original version of this task. In each episode within this environment, $g$ and $s_0$ are uniformly sampled from all possible locations. The agent receives a constant negative reward every step until it reaches $g$. In this experiment, the DQL algorithm is represented as **DRL**, while *goal-reducer* augmented DQL is denoted **DRL+GR**. The results clearly show that **DRL+GR** outperforms **DRL** in terms of convergence time (Fig. 3A right).

**Robot arm reach task**    We adopted panda-gym [19] to implement an environment where a robot arm with 7 degrees of freedom is trained to reach an arbitrary location sampled uniformly in the space (Fig. 3B left). Like the navigation task, the agent receives a constant negative reward every interaction before reaching within a close region centered on the specified goal location. In this experiment, for plain DQL we used SAC (denoted as **DRL**) and *goal-reducer* augmented SAC (**DRL+GR**). The results (Fig. 3B right) are consistent with the navigation task.

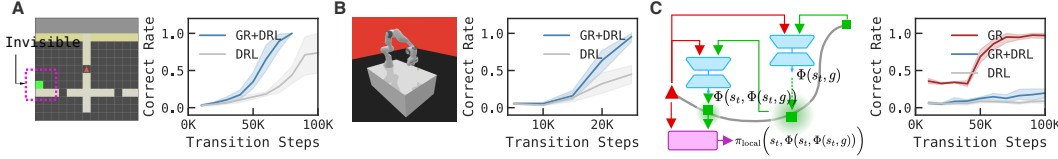

Fig. 3: *goal-reducer* accelerates standard RL. **A**: An example input in the four-room navigation task (left) and performance comparison (right). **B**: Robot arm reach task (left) and performance comparison (right). **C**: How a *goal-reducer* agent works with only a local policy (left) and performance comparison of 3 algorithms (right).

### 4.2 Standalone *goal-reducer* surpasses *goal-reducer*-accelerated RL

The next question we ask is related to the results in recursive goal reduction (Fig. 2C): Can one use just the *goal-reducer* to perform a task as it seems to reduce *Equidex* recursively to neighbor goals? This requires a standalone *goal-reducer* and a "local" policy $\pi_{\text{local}}$ that can learn how to associate $s_t$

and $s_{t+1}$ with $a_t$. The Four-room navigation task naturally fits this need, as a $\pi_{\text{local}}$ in it can be easily defined as a policy that learns to associate two connecting grids. Under this setting, a *goal-reducer* can recursively generate subgoals using existing goals/subgoals until $\pi_{\text{local}}$ finds proper subgoals that are close enough to make a proper decision.

This time a 19x19 maze is used to make the task harder, as *goal-reducer*'s training effect may not be obvious in smaller environments. In this environment, we dropped the Bellman equation (for details, see appendix) and compared its performance (denoted as **GR**) with the previous winner, *goal-reducer* augmented DQL (**GR+DRL**), and the baseline plain DQL (**DRL**). Results in Fig. 3C right column clearly show that **GR** outperforms both **GR+DRL** and **DRL**, while the latter two's performance relationship is consistent with the right column in Fig. 3A.

### 4.3 *goal-reducer* in the Brain

The efficiency the *goal-reducer* has shown in previous experiments, when compared to plain RLs, naturally leads us to wonder if the brain adopts a similar strategy to solve goal-directed behaviors. To address this, we used a cognitive task, treasure-hunting (Fig. 4A), that necessitates flexible goal representation changes. In the task, subjects were placed in one of four possible starting states on one of two maps (Fig. 4B) and were required to reach states designated as a "chest" (the ultimate goal). But having a "key" is necessary when reaching the "chest" to obtain a reward. The locations of the "key" and "chest" are presented to the subject at the start of each episode and change randomly across episodes. The *goal-reducer*, $\Phi(s, g)$, when paired with a $\pi_{\text{local}}$, forms an agent and is also trained on the same task, using the same strategy adopted in Fig. 3C.

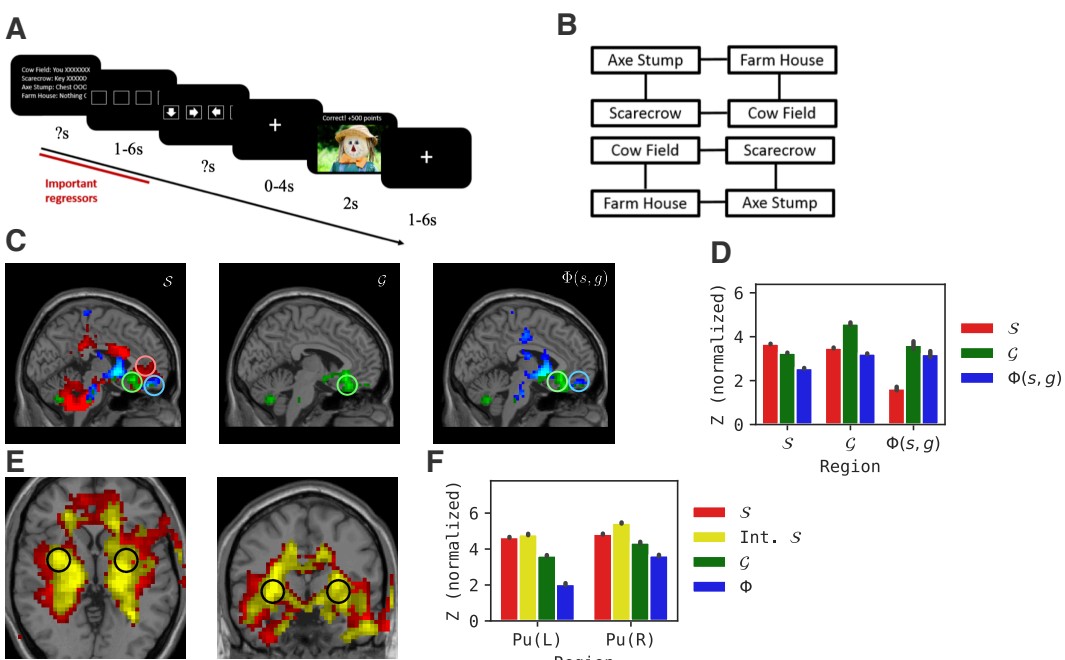

Fig. 4: *goal-reducer* in the treasure hunting task. **A**: The treasure hunting task description. **B**: Two configurations of maps used in the task. **C**: Population z-maps of $\mathcal{S}$, $\mathcal{G}$, and $\Phi$ in vmPFC. **D**: The relative representation z-value distribution for the three centers marked as $\mathcal{S}$, $\mathcal{G}$, and $\Phi$ in vmPFC. **E**: Population z-maps of $\mathcal{S}$, `Int.`$\mathcal{S}$, and $\Phi$ in bilateral putamen. **F**: Same as **D**, but for bilateral putamen.

After training, we analyzed the *goal-reducer* agent's neural activation and compared it with human subjects' brain activity measured via fMRI. The fMRI data, including human subjects' participation details, were reported previously, and human subjects were compensated $25/hr for fMRI participation [54]. In particular, subjects were apprised of the mild risks of fMRI including boredom, fatigue, and loss of confidentiality. All research was approved by the institution's IRB, and all subjects provided full informed consent. Our fMRI analysis used representational similarity analysis (RSA) [32]. RSA considers all conditions that occurred in the task and compares the activity similarity of the

model/brain between each pair of conditions via Pearson correlation, thereby forming a symmetric representational dissimilarity matrix (RDM). The entries in the RDM range from 0 to 2 (1 minus the possible correlation ranging from -1 to 1), where a lower value indicates higher similarity. RDMs are calculated for all voxels in human subjects' brains and for different components in the *goal-reducer* agent, including the input representation $\mathcal{S}$, the goal representation $\mathcal{G}$, and the *goal-reducer*, $\Phi(s, g)$.

The results show that the activity in the ventromedial prefrontal cortex (vmPFC) corresponds to $g \in \mathcal{G}$ in the *goal-reducer* agent. Next, we evaluated the activity matching the internal neurons of the *goal-reducer* $\Phi(s, g)$. As depicted in Fig. 4C, there are several regions whose activities correlate with $\Phi$, including parts of the vmPFC (indicated by the right circle in Fig. 4C) and the left accumbens. This finding is particularly interesting for two reasons: 1) numerous studies have established that the vmPFC is related to value and goal representations [29, 43]; 2) the close spatial relationship between regions matching $\mathcal{G}$ and $\Phi$ (Fig. 4C middle and right columns) reflects the *goal-reducer*'s organization (Fig. 4C right), wherein $\Phi$ and $\mathcal{G}$ are interconnected through the generation and input of different goals. This recurrent connectivity suggests that the widespread recurrent connections in the vmPFC [37] may fulfill the role of goal reduction or perform reverse future simulation [6], thereby simplifying the task. The finding is consistent with a posterior-to-anterior hierarchy of goal planning in the prefrontal cortex [7], and our results may suggest the function of such a hierarchy - namely that anterior regions provide goal reduction functionality for more posterior regions.

Following this, we next investigated the Z map of state representations and indeed found a region above $\Phi$ and $\mathcal{G}$ in the vmPFC that is correlated with $\mathcal{S}$ (as denoted by the top circle in Fig. 4C left). The average z-values in these regions show differential representational loading on the *goal-reducer* agent components (Fig. 4D), although the ROI z-value comparison is a descriptive statistic only given that the regions were selected for strong loading on the various model layers [33].

Aside from the goal reduction, we also compared $\mathcal{S}$ in the *goal-reducer* agent with brain data. The RSA results indicate significantly elevated representational loading for these two layers in the bilateral putamen (Fig. 4E), while there is a lower level of loading for the goal representation layer and an even lesser extent for the goal reduction layer. The putamen, a component of the larger basal ganglia system, is recognized for its involvement in habitual behaviors [53] and goal-directed actions [4, 27]. This suggests that such an area should engage in both local policy enactment (the habitual aspect) and goal reduction (goal-directed component). The congruence between the state representation and its intermediate layer with this role may illuminate why activation related to the goal and its reduction is lower in this region (Fig. 4F), since they are not as essential for the habitual component of the local policy.

## 5   Discussion

**Limitations and future work**    In this work, the trained goal reduction mechanism has shown its capability in terms of its computational advantage and as a biological model of human goal-directed behaviors. This suggests a possible bridge between efficient human problem-solving in multi-goal settings and machine learning. However, a key part of the training of the *goal-reducer*, the loop removal sampling process, does not have a clear mechanistic biological process correspondence. Two ways exist to address this issue in the future. Computationally, it may be possible to derive a purely neural model to perform the loop removal process, making the sampling process also biologically plausible. A potential solution will be to leverage a lateral-inhibition-like [8] mechanism to inhibit all associative synaptic connections between neurons when the same group of neurons are activated more than once in a time range, simulating canceling the "loop" in a memory trajectory. Empirically, it may be possible to collect brain activity data during training of the same or similar tasks, or during offline replay, which may incorporate the loop removal as part of memory consolidation.

**Conclusion**    We developed a novel general goal reduction mechanism using the loop removal trick and trained a network *goal-reducer* that can learn to predict nearly optimal subgoals from distant ultimate goals. We show that this approach does not rely on prior knowledge about the global structure/distance of the environment and uses just random explorations. We further demonstrate that it can be integrated into various existing RL frameworks and outperforms them. Besides, after removing the Bellman equation part, when applied recursively, this framework can perform goal-directed learning and even outperform *goal-reducer* augmented RL methods. This *goal-reducer* agent was next applied to a cognitive task and compared with human fMRI data. Our analyses show

that various regions in the brain, including the vmPFC and basal ganglia, can be mapped to the goal reduction representation and state representations in the *goal-reducer* agent, implying that the brain may instantiate a similar computational circuit to perform goal-directed learning.

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

# A  Appendix

The appendix is divided into four sections. Section A.1 introduces the computational resources used for all experiments. Section A.2 describes the training algorithm used for *Loop-removal sampling*. Section A.3 details the experiment setup and extra results of how *goal-reducer* is used to accelerate RL. Section A.4 is similar to Section A.3, but focuses on the standalone *goal-reducer* experiment. The last section, A.5, includes training and analysis details of the *goal-reducer*-brain fMRI data comparison experiment.

## A.1  Computation resources

We evaluated all simulations on a server with an 11 GB NVIDIA 2080Ti GPU and a computational node with a 32 GB Tesla V100 GPU. The fMRI data processing was executed on a server with 128 GB RAM and an Intel(R) Xeon(R) X7560 CPU.

## A.2  *Loop-removal sampling*

In this experiment, three types of sampling strategies are used. For *Random sampling*, we uniformly sample subgoals from the set of all possible state representations. For *Trajectory sampling*, in a single trajectory, subgoals are uniformly sampled from states that are between the first (start state) and the last state (goal). For *Loop-removal sampling*, we implemented a filtering algorithm to remove loops from trajectories. Below (Algorithm 1) is the pseudocode we used to implement the *Loop-removal sampling* in all experiments:

---
**Algorithm 1** Train *goal-reducer* with *Loop-removal sampling*
---

**Input:** Replay buffer $\mathcal{D}$, filtration threshold $\rho$, and a loop-removed set $\mathcal{D}'$.
**for** each iteration **do**
   Sample a trajectory $d \sim \mathcal{D}$.
   Initialize an empty $d'$.
   **for** each $(s_t, g, a_t, r_t, s_{t+1})$ in $d$ **do**
      **for** each $(s'_t, g, a'_t, r'_t, s_{t'+1})$ in $d'$ **do**
         **if** $||s_t - s'_t|| < \rho$ **then**
            Remove all transitions from $t'$ to the tail of $d'$.
            **break**
         **end if**
      **end for**
      Append $(s_t, g, a_t, r_t, s_{t+1})$ to $d'$.
   **end for**
   Sample $t_0, t_1$ from $d'$ where $t_0 \leq t_1$.
   Append $(s_{t_0}, s_{t_1}, g)$ to $\mathcal{D}'$.
**end for**

---

**Training**   For the two environments in Fig. 2A, we used the same *goal-reducer* architecture for all sampling: a VAE with two architecturally identical 3-layer MLPs for encoding and decoding. The Adam optimizer [30] is used for training. For some important hyperparameters, see Table 1.

Table 1: Hyperparameters used in A.2

| Name | Description | Value |
|------|-------------|-------|
| lr | Learning rate for *goal-reducer* optimizer | $1 \times 10^{-3}$ |
| bsz | Batch size | 256 |
| epochs | Max number of epochs | 4096 |

## A.3  *goal-reducer* accelerates DRL

**Task**   In the four-room navigation task, we created a new environment based on Minigrid [10]. Specifically, we:

- uniformly sampled initial locations and goal locations among all plausible locations.
- changed the action space from a composition of turning angle and moving forward to a plain four directions (up, down, left, right).
- made the environment partially observable to the agent by allowing it to see only a limited squared image of its surroundings (a 13x13 image with the agent in the center).
- made the goal (the green square) invisible to the agent, i.e., the agent will only receive a picture taken as if it is in the goal location as its goal input.

In the robot arm reach task, we adopted the panda-gym library [19]. Initial joint angles and goal coordinates are also uniformly sampled from plausible value ranges.

**Scalability**  One reason we used the four-room navigation task to test the capability of *goal-reducer* is that we can easily adjust the size of the maze. This gives us a sense about how *goal-reducer* with *Loop-removal sampling* scales. As shown in Fig. 3A, a four-room environment is composed of two rooms and three walls (left border, middle, right border) if one looks from a single side, so the legal border size takes the form of $3 + 2k$, where $k$ is the size of the room. We initially tested sizes 15 and 19, as shown in the main text. Next, we examined other sizes and formed results for 13, 15, 17, 19, and 21. From Fig.S. 1, we can see a trend that **DRL+GR** outperforms **DRL** in all cases. Though the gap in a finite step becomes smaller as the sizes increase, given the increased task difficulty, the performance difference is positive and consistent across different conditions.

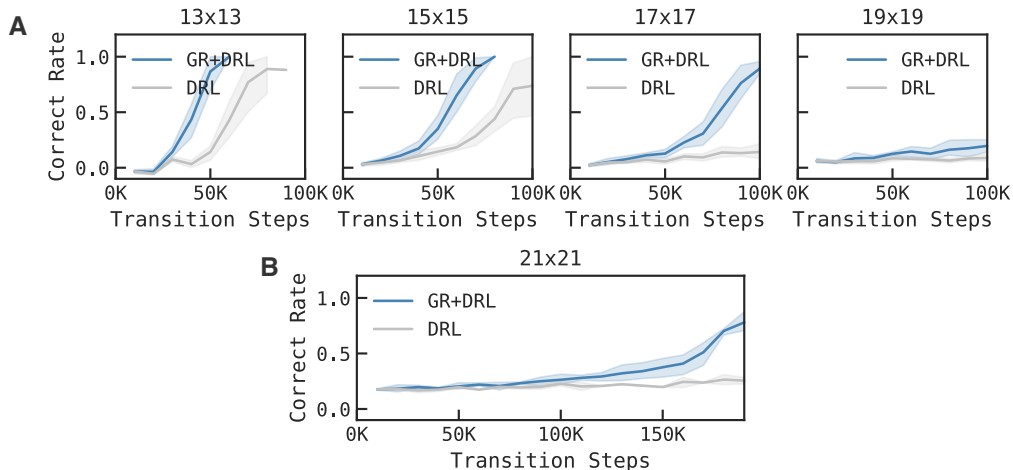

Fig.S. 1: Performance comparison between **DRL+GR** and **DRL** in the four-room navigation task with different sizes.

**Training**  For the **DRL** (DQL) and **DRL+GR** (DQL+*goal-reducer*) agent implementation, we used and modified Tianshou [52] to implement the baseline algorithm and the trainer. To handle pixel-like inputs in the four-room navigation task, an extra single CNN (`ObsEnc`) is used to preprocess the $s_t$ and $g$ (this is not used in the robot arm reach task). Following that, an MLP is used as the Q-net in DQL. Two Adam optimizers are used during training. One is used to train the DQL part, while another is turned on only when the *goal-reducer* is ON. In both **DRL** and **DRL+GR**, we applied the same Hindsight Experience Replay [2] buffer to accelerate training.

For some important hyperparameters used, see Table 2.

### A.4  *goal-reducer* surpasses DRL+GR training details

**Task**  In this experiment, we compared a standalone local policy, **GR**, with **DRL+GR** and **DRL** in a four-room gridworld navigation task.

**Planning process**  In the **GR** case, a standalone *goal-reducer* and a local policy are used to generate actions for every possible $s_t, g$ combination. The pseudocode for this process is listed in Algorithm

Table 2: Hyperparameters used in A.3

| Name | Description | Value |
|------|-------------|-------|
| lr | learning rate for DQL optimizer | $5 \times 10^{-4}$ |
| $lr_{GR}$ | learning rate for *goal-reducer* optimizer | $5 \times 10^{-4}$ |
| bsz | batch size | 256 |
| epochs | max number of epochs | 6 |

2. Using this algorithm, the *goal-reducer* agent can navigate in the environment by recursively producing a batch of subgoals over time, until at least one of them is close enough to the current location. The local policy's entropy plays the role of epistemic uncertainty [56] to help *goal-reducer* decide whether a goal/subgoal is close enough. An example is shown in Fig.S. 2.

---

**Algorithm 2** Action generation of *goal-reducer* with a local policy

---

**Input:** Goal reduction step $t = 0$, max allowed goal reduction steps $K$, state representation $s_t$, goal representation $g$, parallel goal numbers $M$, entropy threshold $\eta$
$a_t = \pi_{\text{local}}(s_t, g)$
**if** Entropy($a_t$) $< \eta$ **then**
    **return** $a_t$
**end if**
Initialize subgoal list $\mathcal{GM} = [g] \times M$
**while** $t < K$ **do**
    **for** $i = 1$ to $M$ **do**
        $\mathcal{GM}[i] = \Phi(s_t, \mathcal{GM}[i])$
    **end for**
    **if** min(Entropy($\mathcal{GM}[i]$)) $< \eta$ **then**
        $g^* = \arg\min(\text{Entropy}(\mathcal{GM}))$
        $a_t = \pi_{\text{local}}(s_t, g^*)$
        **return** $a_t$
    **end if**
    $t = t + 1$
**end while**
**return** $a_t$

---

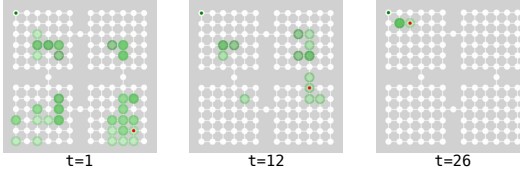

Fig.S. 2: An example of the *goal-reducer* planning process. Red dots show the agent's location, dark green dots (upper left) show the goal, and shadowed green circles show subgoals generated by *goal-reducer* over time. Darker green indicates more subgoals at the same location.

**Training** The training and agent settings are the same as above for **DRL+GR** and **DRL**. For the **GR** case, besides the optimizers for the DQL part and the *goal-reducer* part, we introduced an extra Adam optimizer for the CNN preprocessing part in the pipeline. Without the constraint from a Q function to learn effective state representations, an extra self-supervised learning approach must be used. Similar to [21], we introduced a "world model" (WorldModel) and a decoder (ObsDec) for the agent. Specifically, a decoder and an abstract next state representation are introduced:

$$h_{t+1} = \texttt{WorldModel}(\texttt{ObsEnc}(s_t), a_t) \tag{1}$$

$$\hat{s}_{t+1} = \texttt{ObsDec}(h_{t+1}), \tag{2}$$

where $\hat{s}_{t+1}$ is the prediction of $s_{t+1}$. To train this model, BCE loss between $\hat{s}_{t+1}$ and $s_{t+1}$ is used, paired with a weighted (0.001) L2-norm penalty of $h_{t+1}$ to avoid exploding gradients. For some important hyperparameters used, see Table 3.

Table 3: Hyperparameters used in A.4

| Name | Description | Value |
|------|-------------|-------|
| lr | learning rate for DQL optimizer | $5 \times 10^{-4}$ |
| $lr_{GR}$ | learning rate for *goal-reducer* optimizer | $5 \times 10^{-4}$ |
| $lr_{WorldModel}$ | learning rate for "world model" | $5 \times 10^{-4}$ |
| bsz | batch size | 256 |
| epochs | max number of epochs | 40 |
| M | max goal generation number | 12 |
| $\eta$ | entropy threshold | 0.8 |
| $K$ | max goal reduction steps | 3 |

## A.5 *goal-reducer* fMRI data analysis

**Task** For human subjects, the treasure hunting task is presented with a series of images and verbal instructions on the screen (see Fig. 4A). To train the **GR** agent to perform this task, we turned all presentations into verbal instructions (see Fig.S. 3) and then encoded them with OpenAI's `ada-002` text embedding model to transform them into vectors.

```
==========t=0==========
Goal:
Key: Scarecrow, Chest: Axe Stump

Obs:
You: Cow Field
```

Fig.S. 3: An example of text input in the treasure hunting task for the *goal-reducer* agent

This unifies the representation of this task with all previous tasks and allows future generalization to novel tasks without modifying the agent itself.

**Training** The network architecture is the same as the **GR** agent used in previous experiments. For some important hyperparameters used, see Table 4.

Table 4: Hyperparameters used in A.5

| Name | Description | Value |
|------|-------------|-------|
| lr | Learning rate for DQL optimizer | $1 \times 10^{-3}$ |
| $lr_{GR}$ | Learning rate for *goal-reducer* optimizer | $1 \times 10^{-3}$ |
| $lr_{WorldModel}$ | Learning rate for WorldModel | $1 \times 10^{-3}$ |
| bsz | Batch size | 256 |
| steps | Max number of transition steps | 15000 |
| M | Max goal generation number | 12 |
| $\eta$ | Entropy threshold | 0.8 |
| $K$ | Max goal reduction steps | 3 |

**fMRI data analysis** We included 24 human subjects' fMRI data for analysis.[2] All research involving human subjects was approved by the institution's IRB. Subjects provided informed consent, and the procedures adhered to applicable guidelines and the Declaration of Helsinki. The data were re-used from [54], which also provides the full methods.

---

[2]Data is available at osf.io: part 1, part 2 and part 3.

In the RDM analysis, each RDM for a brain voxel or model component is a $64 \times 64$ matrix, where $64 = 4 \times 8 \times 2$. Here, 4 represents the four possible current subject/model locations on the map. 8 represents all possible episode configurations (comprising four two-step key-chest configurations when the subject/model is at the starting location, two one-step "key-chest" configurations when the subject/model is at the starting location, and two two-step key-chest configurations when the subject/model is at the middle location). The factor 2 accounts for the two phases of each movement (planning and action phase).

For the human fMRI data, such RDMs are calculated for each voxel in the brain for each subject, with a neighborhood radius set to 10 mm. For the model, RDMs are first calculated for learned $\mathcal{S}$ and $\mathcal{G}$ representations and the intermediate representations in the corresponding neural networks. Moreover, we calculated RDMs for the *goal-reducer*'s hidden representations as well when the goal reduction is performed. The action layer, i.e., the output distribution at each step is also calculated. Together, these form the following model RDMs: $\mathcal{S}$ RDM for state representations, `Int.`$\mathcal{S}$ RDM for state representations in hidden layers, $\mathcal{G}$ RDM for goal representations, `Int.`$\mathcal{G}$ RDM for goal representations in hidden layers, $\pi$ RDM for action representation, and $\Phi$ RDM for *goal-reducer*'s hidden layers.

Once RDMs are calculated, Pearson correlation between each voxel and a model RDM's upper triangular part, given their symmetrical nature, is calculated. This forms a series of fMRI RSA maps for each subject, showing whether certain brain regions are positively correlated with the aforementioned model components. Second-level analysis is then performed to extract the population effect. To do so, we first transform the individual correlations into z-scores using the Fisher transform. Then, all z-score maps are projected into a shared space, and a 1-sample T-test is performed for each voxel in the shared brain space to examine whether the corresponding correlation is significant across subjects. After that, cluster-based correction is performed to filter out statistically significant regions. We used nilearn [1], rsatoolbox [39]'s Python version, and SPM5 [42] to perform this process.

With the above analysis pipeline, we found that different brain regions have a high match with distinct components in the *goal-reducer* agent. For a complete list of all clusters, see Table 5.

Table 5: Significant similarity clusters derived from SPM5 [42] for RSA analysis. Anatomical region labels are derived from [36]. The $p_{\text{corrected}}$ and cluster size (cluster$_{k_E}$) values are both corrected using SPM5. Note that some layers may involve more than one region due to the probabilistic nature of atlases in [36].

| Component | X | Y | Z | $p_{\text{corrected}}$ | cluster$_{k_E}$ | Region(s) |
|---|---|---|---|---|---|---|
| Int.$\mathcal{S}$ | 32 | -9 | 21 | 0 | 15,323 | Right Cerebral White Matter |
| Int.$\mathcal{S}$ | 29 | -9 | 11 | 0 | 15,323 | Right Putamen |
| Int.$\mathcal{S}$ | 15 | -12 | -2 | 0 | 15,323 | Right Cerebral White Matter |
| Int.$\mathcal{S}$ | 39 | 43 | 38 | 0.02 | 82 | Frontal Pole, Right Cerebral Cortex |
| Int.$\mathcal{S}$ | -50 | -64 | 28 | 0.01 | 96 | Lateral Occipital Cortex, superior division, Left Cerebral Cortex |
| Int.$\mathcal{S}$ | -44 | -67 | 21 | 0.01 | 96 | Lateral Occipital Cortex, superior division, Left Cerebral Cortex |
| Int.$\mathcal{S}$ | -50 | -47 | 35 | 0.01 | 96 | Supramarginal Gyrus, posterior division, Left Cerebral White Matter |
| Int.$\mathcal{G}$ | -37 | -12 | -26 | 0 | 18,578 | Parahippocampal Gyrus, anterior division, Left Cerebral Cortex |
| Int.$\mathcal{G}$ | -26 | -23 | 1 | 0 | 18,578 | Left Cerebral White Matter |
| Int.$\mathcal{G}$ | -26 | -16 | 32 | 0 | 18,578 | Left Cerebral White Matter |
| $\Phi$ | -30 | -9 | 15 | 0 | 10,362 | Left Cerebral White Matter |
| $\Phi$ | -23 | -26 | 32 | 0 | 10,362 | Left Cerebral White Matter |
| $\Phi$ | 25 | -40 | 21 | 0 | 10,362 | Right Cerebral White Matter |
| $\Phi$ | -13 | 32 | 52 | 0 | 183 | Superior Frontal Gyrus, Left Cerebral Cortex |
| $\Phi$ | -16 | 43 | 45 | 0 | 183 | Frontal Pole, Left Cerebral Cortex |
| $\Phi$ | -13 | 50 | 35 | 0 | 183 | Frontal Pole, Left Cerebral Cortex |
| $\mathcal{S}$ | 29 | -9 | 18 | 0 | 14,266 | Right Cerebral White Matter |
| $\mathcal{S}$ | 25 | -23 | 35 | 0 | 14,266 | Right Cerebral White Matter |
| $\mathcal{S}$ | 18 | -12 | 1 | 0 | 14,266 | Right Cerebral White Matter |
| $\mathcal{S}$ | -50 | -50 | 38 | 0.05 | 70 | Supramarginal Gyrus, posterior division, Left Cerebral Cortex |
| $\mathcal{S}$ | -47 | -54 | 28 | 0.05 | 70 | Angular Gyrus, Left Cerebral White Matter |
| $a$ | 29 | -9 | -13 | 0 | 1,148 | Right Cerebral White Matter |
| $a$ | -2 | -26 | -40 | 0 | 1,148 | Brain-Stem |
| $a$ | 22 | -19 | 32 | 0 | 1,148 | Right Cerebral White Matter |
| $a$ | -16 | 36 | -2 | 0.01 | 104 | Left Cerebral White Matter |
| $a$ | -13 | 32 | -9 | 0.01 | 104 | Frontal Medial Cortex, Left Cerebral White Matter |
| $a$ | -9 | 22 | -13 | 0.01 | 104 | Subcallosal Cortex, Left Cerebral White Matter |
| $\mathcal{G}$ | -16 | 8 | -9 | 0 | 16,195 | Left Putamen |
| $\mathcal{G}$ | 29 | -12 | 28 | 0 | 16,195 | Right Cerebral White Matter |
| $\mathcal{G}$ | 25 | -5 | 8 | 0 | 16,195 | Right Putamen |

