# OpenReview forum: "Goal Reduction with Loop-Removal Accelerates RL and Models Human Brain Activity in Goal-Directed Learning"
_NeurIPS.cc/2024/Conference — NeurIPS 2024 spotlight_

### Official Review · Reviewer_i4LG · 2024-06-30

**Soundness:** 4
**Presentation:** 4
**Contribution:** 4
**Rating:** 9
**Confidence:** 5

**Summary:**

The paper provides a novel way to find subgoals just from past experiences (without any prior knowledge), named goal-reducer. Goal-reducer works by finding “loops” in trajectories and remove them. The authors extend the concept of loops in the continuous cases by defining a filtration radius (distances of two states lower than the radius means they are the same state). Through experiments, the authors show goal-reducer can 1) help RL by providing subgoals; 2) can work on its own to find immediate next subgoals and outperform RL; 3) has a corresponding human brain activity in tasks involving subgoals. In general, the paper is sound, makes meaningful contributions, and is written well.

**Strengths:**

1. The writing is very easy to follow and describes contributions very clearly w.r.t. previous methods.
2. The paper looks into an important problem in both computer science and neuroscience, finding connections between the two mechanisms. The problem is motivated well in introduction.
3. The proposed method is intuitive yet very effective. It makes sense both theoretically and empirically via experiments. The proposed method also removes the need for prior knowledge about the environment and the task, which are required by most previous methods.
4. The experiments support the paper’s claims well. I appreciate the authors’ effort in removing domain knowledge by random embedding for raw states on Line 200.

**Weaknesses:**

1. I am not sure about the applicability of goal-reducer beyond just goal-reaching behavior. Take the fMRI study task as an example, how do we make sure the goal-reducer does not condense these two states together: both states are in the same place, but one with the “key” and one without the “key”. If the mapping to embedding space is random, we cannot guarantee the distance of having/not-having the key is larger than the distance of being in two different places. Essentially, I am saying that what if the loop detour is necessary to get some treasure. I wonder about the authors’ thoughts of the applicability of goal-reducer in harder tasks than just navigation.
2. I am not sure about the motivation/justification for Equation 6-7: why should the loss be present only when the entropy decreases when the subgoal is introduced? Wouldn’t that encourage the policy to be less certain about what to do when given the subgoal?
3. Line 297 mentioned Indiana University IRB – while it is great to show the study was IRB-approved, I wonder if this is in violation to anonymity requirements in reviewing stage.

**Questions:**

See weakness

**Limitations:**

While the paper discusses their approach's limitations, they do not discuss the potential negative societal impact of their work. I hope the authors consider including them in the rebuttal.

---

> ### Author Rebuttal · Authors · 2024-08-05
>
> We appreciate the feedback and valuable insights provided by you. For your questions, here are some of our thoughts:
>
> 1. Applicability issue: We agree that when a detour is necessary, things could be confusing. However, to our knowledge this usually depends on the problem formulation. For example, when a certain detour, e.g., capturing a distant key is a prerequisite to unlock the reward in the box, the goal space will no longer be the location space, and it will be occupancy of both key and box. With such transformations, all tasks can be formulated in the form that is solvable by this approach.
> 2. Eq 6 and 7 issue: Eq. 6 and 7 help to train the Q-net. The process relies on two crucial assumptions: First, a Q net will have larger entropy when it’s not sure about its own prediction. Second, “closer” goals are easier to learn (and thus have lower entropy) for a Q-net. Third, a goal reducer may work independent of the Bellman equation. With these assumptions, Eq. 7 says that when a Q-net is not very sure about its prediction (larger entropy) of the ultimate goal, it should trust a closer subgoal more (lower entropy), as the subgoal is generated by an independent goal reducer (assumption 1 and 3). And since a closer subgoal’s Q-value is easier to learn and it’s informative about the ultimate goal due to the recursive nature, we can use the Q-value distribution to regularize the ultimate goal’s Q-value distribution.
> 1. IRB issue: We apologize for identifying the  IRB  of record. This will be changed to “institution’s IRB” in the final version.

---

> > ### Comment · Reviewer_i4LG · 2024-08-11
> >
> > Thank you to the authors for answering my questions! I will keep my score.

---

### Official Review · Reviewer_SziM · 2024-07-12

**Soundness:** 2
**Presentation:** 2
**Contribution:** 2
**Rating:** 4
**Confidence:** 5

**Summary:**

The paper proposes a goal reduction mechanism utilizing a novel loop-removal technique to derive subgoals from arbitrary and distant original goals. This method, called goal-reducer, distills high-quality subgoals from a replay buffer without prior global environmental knowledge. It accelerates performance in various RL frameworks and matches human brain activity in goal-directed tasks.

**Strengths:**

- This study presents an innovative method for eliminating loops in order to reduce goals, hence condensing intricate goals into more manageable subgoals. This breakthrough tackles the issue of extensive state and goal spaces in conventional RL algorithms, enabling faster and more effective learning.
- The goal-reducer can be easily integrated with existing RL frameworks like DQN/SAC, enhancing their performance without requiring major modifications to the underlying algorithms

**Weaknesses:**

Refer to questions.

**Questions:**

- In algorithm 1, how to calculate $||s_t - s_t'||$? Does this distance depend on Euclidean distance? If yes, the Euclidean distance is a strong prior knowledge of the environment to the agent.
- I agree that SoRB relies on strong prior knowledge. However, there are some related work that trains a sub-goal generator along with the RL learning process, e.g., [1].
- Can this method be extended to any goal state? If there are multiple goals or no specific goal state is specified, then removing the historical state traversed by the agent will weaken the agent's ability to reach certain goals.
- Why didn't the authors compare the performance of GR+DRL  with SoRB?

[1] Shuang Ao, Tianyi Zhou, Guodong Long, Qinghua Lu, Liming Zhu, Jing Jiang, CO-PILOT: COllaborative Planning and reInforcement Learning On sub-Task curriculum

---

> ### Author Rebuttal · Authors · 2024-08-05
>
> Thank you for the insightful feedback as well as pointing out pieces we missed in the original writing. Regarding your questions,
>
> 1. State distance estimate issue: Yes we used Euclidean distance to measure the similarity between different states representations, however, this does not necessarily impose prior knowledge during training. As we’ve shown in line 201, the state representations are random vectors without any knowledge about the actual cost between different states. But definitely, if we use a trained observation encoder, the performance will be better.
> 1. Missing references issue: We apologize for the unintentional omission of Shuang et al., which we now cite and discuss as prior work in the 2nd paragraph of section 2.1 and this will be reflected in the final version.
> 1. Arbitrary goal issue: We apologize that the model’s strengths were not described more clearly. The model method can be extended to any goal state as well as underconstrained goal sets. It must be noted that the goal space is not necessarily the state space. For example, if a task requires an agent to collect $K$ gemstones to get a reward, the goal state is defined as having obtained all $K$ gemstones and correspondingly a subgoal might be having obtained  $K-1$ gemstones. With this formulation, the loop removal trick still works.
> 1. SoRB issue: We agree comparing SoRB with goal reducer will be an extremely important and intriguing  question to explore in our future work. Because SoRB has one huge advantage and also a potential drawback: it explicitly builds a graph and thus the search will be mathematically optimal. However, the quality of its state graph relies on the quality of the value function, this may make the process ineffective in the early stage of training, when the Q net is still not mature enough. In contrast, our model can work with less prior knowledge in estimating the subgoal (or waypoint in SoRB) while we suffer from a trained goal reducer, who may not always generate the optimal subgoal. So this could result in a crossing effect: Our model may converge faster initially and have an overall a bit lower performance when compared with SoRB after long enough training.

---

> > ### Comment · Reviewer_SziM · 2024-08-11
> > **Response**
> >
> > Thank you for the author's reply, which solved some of my problems. However, I still stick to my point of view that Euclidean distance is a strong prior knowledge, which is a critical ability of the agent. I agree that for the agent, the size of the Euclidean distance is not strictly equal to the cost, but imagine if we have the distance information, the goal generator is essentially a search problem. SORB does not rely on this prior knowledge, which explains why SORB might be inefficient in the early days when the Q network was immature.

---

> > > ### Author Response · Authors · 2024-08-13
> > >
> > > We thank the reviewer for the feedback. We are concerned, however, that the use of the Euclidean distance calculation in our work may have been misunderstood. When the agent is randomly initialized, there is no prior knowledge other than that **repeatedly visited states share the same or very similar hidden representation and thus imply the existence of a loop**. This is a minimal and universal assumption across different tasks. Beyond recognizing “loops”, we do not use Euclidean distance for any other purposes, such as value/cost estimation.

---

### Official Review · Reviewer_ww9y · 2024-07-13

**Soundness:** 4
**Presentation:** 3
**Contribution:** 4
**Rating:** 7
**Confidence:** 2

**Summary:**

This paper introduces a new automated method to identify subgoals in reinforcement learning. The method is based on a replay buffer of agent trajectories and proceeds by identifying and eliminating “loops” (points where the agents revisit almost the same state), in effect attempting to identify efficient shortcuts in the replays and set those shortcuts as subgoals. This method for subgoal extraction can be applied to a variety of reinforcement learning algorithm, and is successfully demonstrated with two popular approaches (deep Q learning and soft actor critic), leading to performance gains in two environments (maze exploration and robotic arm control). In addition, a human neuroimaging experiment shows that the latent space of the “goal-reducer” components of an agent shares similarity with brain activity in key regions, suggesting that the proposed goal reduction mechanism may be at play in biological networks. In summary, the paper proposed a new broadly applicable approach to identify subgoals for reinforcement learning, demonstrates its effectiveness on established RL benchmarks and also offers evidence in support of a biological implementation of this process in human brains The paper thus presents a very strong and multidisciplinary set of evidence.

**Strengths:**

The paper is well written. I am not an expert in RL and found it hard to follow at times, with heavy use of mathematical notations without qualitative explanations (the blocks of equations 4 to 7 for DQN for example). Still, I found the algorithm well motivated and the analogy with Dikjstra’s shortest path algorithm helped me understand the motivation of local optimization well I think.

The fact that the method can be applied across a range of RL approaches, or stand-alone, is appealing.

Combining a novel method, a rigorous RL benchmark with a complex set of human neuroimaging experiments in one paper is impressive.

The methods, algorithms and experiments are very carefully described, including the version of software used and details on the hardware used to run the experiments, aiding reproducibility.

**Weaknesses:**

Although the authors claim that the code was shared with the manuscript, I was not able to find it. I either did not look in the right place (I was expecting a supplementary material link on the submission page, or a link in the manuscript), or the authors forgot to upload it. In any case, access to the code for review would address this weakness, but in the current state I am not able to assess the quality of the code and whether the code seems to cover all the parts in the paper. Also, it is stated that the fMRI data is available “upon reasonable request”, it is not possible for me to assess to what extent the authors will actually be responsive, and whether the data is reusable. I strongly encourage the authors to use an established open data sharing platform such as openneuro and standard data structure such as BIDS to increase reproducibility. Statistical maps could be shared on neurovault for increased reproducibility.

It would have been beneficial to contrast different models in the neuroimaging experiment, and incorporate some baseline models and an ablation study. The origins of the similarity in representations may be trivial.

**Questions:**

The authors extend the grid exploration task to 19 x 19 cells in order to test the scalability of the method. Why 19? And not 20? Have the authors found a “breaking point” in the scalability of the approach? Also, insights on the computational scalability of the approach would be useful.

The authors identified similarity of representation with the goal reducer in the white matter (which appears significant in the provided putamen slices). How do the authors explain that? Could it suggest some motion or physiological artifacts correlated with the artificial representations?

Although fMRI results are presented, this neuroimaging modality has very limited temporal resolution. Could this affect the representational similarity analysis? It looks like the steps of the games are slow by design, but this could be discussed briefly.

**Limitations:**

There is no clear neuronal mechanism proposed in support of the goal reduction process, beyond association with an indirect brain measure (blood oxygenation). But bringing any kind of multidisciplinary evidence is already very commendable.

---

> ### Author Rebuttal · Authors · 2024-08-05
>
> Thank you for your detailed and constructive feedback. Regarding equation 4-5, they represent the Bellman part in RL, equation 6 and 7, together, shows how a goal reducer can accelerate RL by using the generated subgoal to regularize (KL divergence is used) the learning of value of the original goal. Specifically, when the entropy of a Q net towards an ultimate goal state is higher than that of a subgoal generated by the goal-reducer, we use the Q value of the subgoal to teach the ultimate goal’s Q value distribution. This relies on two assumptions: 1) “closer” goals are easier to learn than distant goals in RL and 2) the goal-reducer can generate fine subgoals.
>
> Regarding the code, previously we put the code in a github repo, but due to the NeurIPS requirement of anonymity, we removed the corresponding URL in the final submission as it would have identified us. The same applied to the fMRI data, which we published on osf.io. These changes will be reflected in the final version.
>
> Regarding your questions,
>
> 1. Size issue: 19 is used because we adopted a 4 room setting in the navigation task, where in the center we need a “wall”, so to make all rooms have the same size, i.e, 1 (left wall)+ 8 (left room) + 1 (center wall) + 8 (right room) + 1 (right wall). Due to limited computation resources, we didn’t try environments with larger sizes. However, we observed consistent trends for sizes like 11, and 13, 15, 17, and 19, though as the sizes increase, all models converge more slowly. After we received your feedback, we’ve run experiments with size 21 and the same trend persists, namely that performance with the goal reducer is superior. In the future, one direction to go will be to scale the model on larger environments with more powerful computational resources. The latest size 21 result will also be updated in the supplementary material.
> 1. White matter issue: In our fMRI analysis, we found very widespread state representation across the brain. We see white matter due to the putamen's close proximity to white matter tracts. The 3D smoothing process may contribute to the spillover effects in the white matter.
> 1. Temporal resolution issue: Yes, in order to capture the existence of the potential subgoal reduction process, the steps of the games are in fact slow by design. Moreover, there are several events within each trial, and we included jitter in the timing between events in order to estimate the BOLD response to each event within a trial separately, as is common practice. These individual event beta weights are in turn individually present in the RSA matrices, so that the activity of each event within a trial is accounted for. These events are included for each of the trial conditions which include various combinations of state and goal pairs. Thus the RSA results account for events within trials as well as the various task conditions tested.
>
> Lastly, we thank the reviewer for inquiring about potential neuronal mechanisms of goal reduction in the brain. The goal reduction process involves two key components: subgoal generation and loop removal. The subgoal generation is already "neural" as we used a network to generate subgoals rather than other heuristics. Regarding loop removal, our hypothesis is that feedforward inhibition in the hippocampus may implement it. Consider a replay event in which the animal has already traversed a point twice (a loop); the average activation of that place cell (PC) pattern will be stronger than other PCs. If a threshold is set between them, an inhibition mechanism may inhibit all PC patterns between the two activations, thus strengthening non-overlapping PC associations while suppressing "loops". Of course, more refined simulations and experiments are necessary to validate this.

---

> > ### Comment · Reviewer_ww9y · 2024-08-12
> >
> > Thanks for the detailed explanations on the equation. This was more of a general comment, but I still appreciate the effort.
> >
> > Re code and reproducibility, I have received a note below that the code was shared, and meets acceptable standards of quality.
> >
> > Re Q1 (size of room). Thanks for the explanations. Scaling the experiment as was done by the authors in response to reviews further strengthens the conclusions.
> >
> > Re Q2, it's indeed possible that white matter activations are an effect of smoothing. This could be tested by reducing smoothing (although it's not advisable for the main analysis: smoothing is required for improving SNR and correction of multiple comparisons)..
> >
> > Re Q3: thanks for the clarifications.
> >
> > Also thanks for the speculations regarding the putative role of the hippocampal formation in loop suppression.
> >
> > I am keeping the score at 7 "accept".

---

> ### Comment · Area_Chair_QDgd · 2024-08-06
> **Code**
>
> Dear Reviewer,
>
> The authors have privately send me a link to their code and data for the human studies (in line with NeurIPS policies about sharing code). The data takes the form of several `.mat` files. The code seems reasonable and contains documentation for installation and for running experiments. During the discussion period, please respond about whether this addresses your concerns about the code and data.
>
>
> Kind regards,
>
> The AC

---

> > ### Comment · Reviewer_ww9y · 2024-08-12
> >
> > Thanks for this information!

---

### Author Rebuttal · Authors · 2024-08-06

We appreciate the reviewers' thoughtful feedback and the opportunity to clarify several key aspects of our work. Individual rebuttals are made below to avoid confusion. In this global rebuttal, we aim to address the concerns raised by reviewer [ww9y](https://openreview.net/forum?id=Y0EfJJeb4V&noteId=vRAdVDDOqj), which pertain to the scalability of the goal reduction mechanism. In short, we show that the trend of goal reduction outperforming its deep RL counterparts is consistent across different environment sizes and persists even beyond the range we tested in the original submission. However, the computation resources have limited us from running even larger tests. This will be a direction to explore in our future work. For details, see the uploaded PDF.

---

### Decision · Program_Chairs · 2024-09-25

**Decision:**

Accept (spotlight)

**Comment:**

This paper proposes a reinforcement learning method (snipping out loops in collected data), presents results showing that this method improves performance, and shows evidence that this method may be similar to biological networks. Reviewers found the paper well written, commending the paper for including both RL experiments and human neuroimaging experiments. Reviewers also appreciated the high degree of reproducibility, experiments demonstrating that the proposed method works for multiple RL backbones, and the discussion about removing the need for domain knowledge. The reviewers asked some conceptual questions that the authors clarified during the review process. The authors shared code with the AC, per NeurIPS guidelines. During the reviewer discussion, Reviewer i4LG (score=9) reviewed the concerns of Reviewer Szim (score=4), and agrees with the authors that those concerns may stem from a misunderstanding. Taking this all into account, I recommend that the paper be accepted.